# Carprofen Permeation Test through Porcine Ex Vivo Mucous Membranes and Ophthalmic Tissues for Tolerability Assessments: Validation and Histological Study

**DOI:** 10.3390/vetsci7040152

**Published:** 2020-10-10

**Authors:** Lidia Gómez-Segura, Alexander Parra, Ana C. Calpena, Álvaro Gimeno, Antonio Boix-Montañes

**Affiliations:** 1Department of Medicine and Animal Health, Faculty of Veterinary, Autonomous University of Barcelona, Bellaterra, 08193 Barcelona, Spain; lidia.gose@gmail.com; 2Department of Veterinary Medicine and Zootechnic, Faculty of Agricultural Sciences, University of Applied and Environmental Sciences, RX22+57 Bogota, Colombia; aleparra@udca.edu.co; 3Department of Pharmacy and Pharmaceutical Technology and Physical Chemistry, Faculty of Pharmacy and Food Sciences, University of Barcelona, 08028 Barcelona, Spain; anacalpena@ub.edu; 4Institute of Nanoscience and Nanotechnology (IN2UB), University of Barcelona, 08028 Barcelona, Spain; 5Department of Animal Research, Animal House of Bellvitge, University of Barcelona, CCiT-UB, Hospitalet de Llobregat, 08907 Barcelona, Spain; alvarogimeno@ub.edu

**Keywords:** carprofen, HPLC, permeation test, histological integrity, mucous tolerability, ophthalmic tolerability

## Abstract

Carprofen (CP), a non-steroidal anti-inflammatory drug (NSAID), is profusely used in veterinary medicine for its analgesic and anti-inflammatory activity. Some undesirable effects are associated with its systemic administration. Alternative local routes are especially useful to facilitate its administration in animals. The main aim of this paper is to validate the suitability of ex vivo permeation experiments of CP with porcine mucous membranes (buccal, sublingual and vaginal) and ophthalmic tissues (cornea, sclera and conjunctiva) intended to be representative of naïve in vivo conditions. Chromatographic analysis of CP in membrane-permeated samples and drug-retained have been validated following standard bioanalytical guidelines. Then, recovery levels of drugs in tissue samples were assessed with aqueous phosphate buffered saline (PBS) buffer to preserve the histological integrity. Finally, as a proof of concept, a series of CP permeation tests in vertical Franz diffusion cells has been performed to evaluate permeation flux and permeability constants in all tissues, followed by a histological study for critical evaluation. Furthermore, synthetic tissue retention-like samples were prepared to verify the value of this experimental study. Results show linear relationships with good determination coefficient (R^2^ > 0.998 and R^2^ > 0.999) in the range of 0.78 to 6.25 mg/mL and 3.125 mg/mL to 100 mg/mL, respectively. Low limits of quantification around 0.40 µg/mL were allowed to follow permeation levels until a minimum of 0.40% of the locally-applied dose. This method showed a good accuracy and precision with values lower than 2%. After the recovery technique, reproducible values below 30% were achieved in all tissues, suggesting it is a non-damaging method with low efficiency that requires the use of further solvents to enhance the extraction percentages. After permeation and histology tests, no relevant peak interferences were detected, and no cell or tissue damage was found in any tissue. In conclusion, results demonstrate the suitability of this test to quantify the distribution of CP with good histological tolerability.

## 1. Introduction

Carprofen (CP) is a non-steroidal anti-inflammatory drug (NSAID) that inhibits microsomal cyclooxygenases (COX), which catalyze the oxygenation of arachidonic acid to prostaglandin H2 (PGH2) [1]. Type 2 cyclooxygenase (COX-2) is the inducible form of the enzyme and is related with inflammation processes.

Currently, CP is not used in humans due to episodic adverse effects such as photoallergic cases after mishandling on an industrial scale [2]. However, it is a drug widely used in veterinary medicine as an anti-inflammatory. There is variety of veterinary therapeutic indications: osteoarthritis [3,4], bovine mastitis [5], analgesic [6,7] and respiratory diseases requiring antibiotics [8,9]. Its absorption kinetics have been investigated in numerous species of animals (horse [10,11], dog [12,13,14], cat [15], broiler chicken [16] or rat [17]). However, to the extent of our knowledge, there are no studies in pigs. Its chronic administration is not preferable for its chirality [18] and its systemic distribution is associated with gastric ulcers, abdominal pain, vomiting and nephrotoxicity [19]. If administered systemically and over a long period of time, it has been reported to cause occasional liver toxicity [20,21,22].

This hepatoxicity, detectable with histopathologic documentation of hepatic necrosis and also renal tubular disease, requires its discontinuation [23]. Additionally, episodes of NSAID phototoxicity [24] can be avoided if systemic distribution is precluded. For these reasons, it is expectable that its local administration would provide a better tolerability for both humans and animals as with other NSAIDs [25], although its pharmacokinetics is still not well known. A successful local administration ensures adequate drug amounts at the site of action at a suitable rate. Therefore, the evaluation of its transmembrane permeability and retention with ex vivo membranes demands its analytical quantification in the target tissues to assure drug level profiles in these series of tissues, for example, mucosal or ophthalmic tissues.

Thus, the objective of this investigation was to validate a quantification method of CP in permeation and/or membrane retention samples following standard bioanalytical and permeation test guidelines [26,27,28,29], and using different porcine membranes, obtained ex vivo, available for local administration such as cornea, conjunctiva and sclera and also sublingual buccal and vaginal mucous membranes.

As a proof of concept of this preliminary validation, the intrinsic permeation of CP through these tissues was investigated to evaluate the adequacy of this ex vivo test to estimate the corresponding parameters of drug permeation–penetration. In addition, we confirm there is no damage toa cell structure with a histological examination.

## 2. Materials and Methods

### 2.1. Chemicals and Reagents

CP was supplied by Capot Chemical (Hangzhou, China), deionized water (18.2 MΩ.cm) was generated with a MilliQ System (Waters corp.). Potassium dihydrogen phosphate (H_2_KPO_4_), disodium phosphate (HNa_2_PO_4_) and methanol analytical grade were supplied by Panreac Quimica (Barcelona, Spain) and dimethyl sulfoxide (DMSO) was acquired from Fisher Scientific (Fisher Scientific, Leicestershire, UK). All other chemicals and reagents had suitable analytical grades (Fisher Scientific, Leicestershire, UK).

### 2.2. Biological Materials

Specimens of mucosal and ocular tissues were obtained ex vivo from female pigs (Landrace x Large White cross, 40–45 kg), resulting from surplus of surgical studies following the study protocol of the Animal Experimentation Ethics Committee of the University of Barcelona with ethic code 514/18. Animals were anesthetized with intramuscular (i.m.) ketamine (3 mg/kg), xylazine (2.5 mg/kg), and midazolam (0.17 mg/kg) and, after the surgery practices, pigs received an intravenous (i.v.) overdose of sodium thiopental. Immediately, mucous membranes, buccal, sublingual and vagina, and eye balls, were excised, debrided in the laboratory to obtain the corresponding sclera, cornea and conjunctiva and frozen by placing them in containers with a phosphate buffered saline (PBS) mixture containing 4% albumin and 10% DMSO (as cryoprotective agents) and stored at −80 °C in a mechanical freezer until their use (see Figure 1). The day before their use, specimens were cut in 700 µm parallel slices with a mucotome GA 630 (Aesculap, Tuttlingen, Germany) [30,31,32] (see Figure 1 and Figure 2) and mounted in vertical Franz-type cells of diffusion (Vidra Foc, Barcelona, Spain) to perform the ex vivo experiment. In addition, other untreated samples were also saved to perform the histological study later.

In these experiments of diffusion (described later in the text), two kinds of samples were expected to be obtained: permeation samples consisting of CP solutions in buffer saline, and penetration samples, consisting of samples of each tissue that imbibed CP (drug retention), which required an additional extraction procedure prior to the HPLC analysis.

### 2.3. Buffers and Solutions

Phosphate buffered saline (PBS) was formulated dissolving HNa_2_PO_4_ (11.88 g) and H_2_KPO_4_ (9.08 g) in 1 L deionized water (20 mM). pH was adjusted to pH = 7.4 using a pH meter (Crison Instruments S.A., Alella, Spain) correcting the osmolality to 300 mOsm/L aided with a Fiske 3320 osmometer (Advanced Instruments, Norwood, MA, USA) and stored at 5 °C for a maximum of three months.

Mobile phase for ultra-violet high-performance liquid chromatography (HPLC-UV) analysis was prepared by dissolving H_2_NaPO_4_ (1.36 g) in 1 L of deionized water, adjusting to pH = 3.0 using a pH-Meter (Crison Instruments S.A., Alella, Spain) and diluting 25:75 with methanol.

pH 7.4 buffer phosphates saturated with tissue (PBSm: PBS saturated with mucous and PBSo: saturated with ophthalmic tissue) were made by immersing freshly made PBS in minced mucous or ophthalmic tissue specimens (1:100, *w*/*w*) during 6 h continuous agitation. They were used to prepare the dilutions of calibrator curve standards for mucous membranes (PBSm) or ophthalmic tissues (PBSo).

CP saturated solution 1500 µg/mL (CP-Sat) was prepared adding to a glass beaker 100 mL PBS and small aliquots of CP towards 150 mg (near saturation) under stirring. After 10 min repose, solution was filtered through a 0.45 µm nylon filter (Teknokroma, Barcelona, Spain) and stored at 20 °C.

### 2.4. Calibration Curve Standards (CCS), Quality Control Standards (QCS) and Fortified Samples (FS)

Two series of calibration standards were prepared with respective PBSm or PBSo imitating the samples of mucosal or ophthalmic permeation tests. Two calibration ranges were established, depending on the expectable experimental concentrations: 0.78 to 6.25 and 3.125 to 100 µg/mL. Mother solution was prepared daily as a 1000 µg/mL stock solution. Dilutions were prepared diluting the corresponding aliquots in PBSm or PBSo in 10 mL volumetric flasks and stored at 5 °C.

For the analytical runs of real samples, quality controls (100, 6.25 and 0.78 µg/mL) were prepared and stored at 5 °C (for no more than ten days) to be used as interleave samples in each run: initial, middle, final of each sequence. Sample run was considered acceptable if relative errors of QCS were lower than ±5%.

Tissue samples fortified with CP [29] were prepared for the evaluation of drug retention recovery: 1 mL CCS-100 (100 µg/mL) was added to accurately weighed samples of each tissue (in duplicate). Each vial remained at 37 °C/6 h to simulate the membranes during a permeation experiment. Afterwards, the liquid was removed, the membranes rinsed, and each series of liquids were stored in vials to analyze the residual amount of carprofen. The difference with the initial amount of carprofen was considered to be the amount of drug charged inside these fortified samples.

### 2.5. Carprofen Solubility

Aliquots of 10 mg CP were iteratively incorporated in 100 mL fresh PBS with continuous stirring until transparency disappeared (true saturation). Subsequently, the vial was stored for 24 h/20 °C in darkness to become limpid. The supernatant was filtered through nylon 0.45 µm and diluted 1:1000. Finally, it was analyzed with HPLC-UV and interpolated in a calibration line yielding a CP solubility of 1781.92 µg/mL.

### 2.6. HPLC-UV

Chromatographic analysis (Waters LCM1 plus; Waters Co., Milford, Maryland, USA) was performed with a C18 reversed-phase column (Simmetry 5 µm 3.9 × 150 mm^®^, Waters Co., Dublin, Ireland). The mobile phase was filtered through a 0.45 µm nylon membrane (Technokroma, Barcelona, Spain) and degassed by sonication. Calibration standards and permeation samples were stored at 5 °C in closed vials that contained a minimum of 50 µL of each sample. Drug was eluted with a flow rate of 1.0 mL/min, detected at 235 nm inside the range of 2 to 3 min and integrated with serial Millenium^®^ software (Waters Co., Milford, MA, USA).

### 2.7. Analytical Method Validation

Validation of the analytical procedure consisted of the evaluation of linearity, range, accuracy, recovery and precision following the concerned indications from the guidelines of European Medicines Agency [26], Asociación Española de Farmacéuticos de la Industria [27], Conference on Harmonization guidelines (ICH) [28] and the Scientific Working Group for Forensic Toxicology [29]. Results were treated considering two calibration ranges: from 100 to 3.125 µg/mL in a high concentration level and from 6.25 to 0.78 µg/mL in a lower concentration level.

#### 2.7.1. Specificity–Selectivity

It was assessed using actual samples from preliminary mucous or ophthalmic tissues permeation experiments generated as described in Section 2.2. Samples were blanks of PBSm or PBSo (*n* = 3) and blanks of PBSm or PBSo spiked with CP at 6.25 µg/mL (*n* = 3) stored in closed vials and exposed to 37 °C for 6 h prior to quantification. In addition, three last-time samples of tissue permeation tests (mucous samples and ophthalmic tissues) were also analyzed.

Resolution between CP and the first adjacent peak was estimated as described in previous studies [33]: peaks of mobile phase or sample baseline should not interfere with active peak. The peak of CP is pure (purity threshold > purity angle) and the resolution between adjacent peaks is greater than 1.5.

Assuming that a source of unspecificity could be associated with any alteration of the histology, membranes used in the tissue permeation test were inspected with the microscope as described in Section 2.8.4.

#### 2.7.2. Linearity and Range

Linearity was verified from the assessment of precision and accuracy (see Section 2.7.4) and calculated with calibration standards prepared with PBSm (from CCS 0.78 to CCS 100) in three interday runs. The least square linear regression analysis and mathematical calculations were performed with Prism 5.0 software (GraphPad software Inc., San Diego, CA, USA). The slopes between the instrumental signal and the nominal drug concentration were estimated and differences as a function of the concentration level were evaluated with a one-way ANOVA. Differences were considered statistically significant if *p* < 0.05.

The working range between the lower and upper concentrations [28] was assumed if determination coefficients were significant (*p* < 0.05) and precision and accuracy were considered adequate as described in Section 2.7.4.

#### 2.7.3. Determination Limits

Limits of detection (LOD) and quantification (LOQ) were calculated (*n* = 3) based on the standard deviation of the response and the slope of the calibration curve, used as described formerly [27,33].

#### 2.7.4. Accuracy and Precision

Precision and accuracy values were calculated at the concentration levels of: CCS-100, CCS-6.25 and CCS-0.78. The interday values were calculated after analyzing the samples of 3 different days.

Precision was expressed as the relative standard derivation (RSD, %). The accuracy, being the closeness between the interpolated value and the nominal concentration of the standard [34], was expressed as the maximum values of the relative error (RE, %). Both were calculated as described in documents cited under Section 2.7.

#### 2.7.5. Stability of Standards

Samples obtained during the permeation experiments were stored at 5 °C and daily analyzed without intermediate freezing. No sample stability for permeation experiments was investigated. Stability of CCS and QCS standards was verified by storing samples at 5 °C and comparing results of area/concentration ratios at the beginning and at the end of the intended storage times with an ANOVA analysis.

### 2.8. Applicability of the Method

In addition to the proper validation of the chromatographic quantification of CP in both types of samples, a series of permeation experiments [35] was run as a proof of concept of the experimental setup. Main premises to be fulfilled were sink conditions across the transmembrane flux; likelihood of the drug retention levels and histological integrity of the membranes after the experiment.

#### 2.8.1. Permeation Test with Diffusion Cells

Ex vivo CP permeation test was run with vertical Franz diffusion cells (Vidra Foc, Barcelona, Spain). Surface area was 0.64 cm^2^ and receptor chamber capacity was 4.5 mL for all the mucous membranes except for buccal mucous membranes (2.54 cm^2^ and 12 mL of receptor chamber capacity) (see Figure 2).

Membrane specimens (mucosal or ophthalmic) were placed with the external side in contact with the donor compartment and the proximal side facing the receptor side. Then, 300 µL of CP-Sat were sown in the donor compartment. PBS was used as receptor medium under continuous stirring (200 rpm) for homogenization and kept at 37 ± 0.5 °C, except for cornea (32 ± 0.5 °C).

After slowly defrosting, mucous membranes were placed in respective Franz diffusion cells. Then, 300 µL CP-Sat was placed in the donor compartment and covered with Parafilm^®^ to avoid evaporation. Samples of receptor compartment (300 µL) were withdrawn at pre-defined times during the 6 h and replaced with an equivalent volume of fresh PBS at the same temperature. CP in samples was quantitated as described in Section 2.6.

The cumulative CP permeated amounts (µg) at each time (min) were calculated from the concentration of CP in the receptor medium.

#### 2.8.2. Recovery from Porcine Mucous Membranes or Ophthalmic Tissues

Quantification of CP retained inside the membranes required an extraction prior to analysis.

The sample series of fortified samples (FS*) were submitted to extraction as follows. Firstly, they were superficially cleaned with gauze soaked in a 0.05% solution of dodecyl sulphate followed by distilled water. Mucous or ophthalmic tissues were perforated using a 30 G needle (BD Ultra FineTM, Beckton Dickinson, Fraga, Spain), minced carefully and weighed accurately. The CP content was extracted with PBS under sonication (20 min) in an ultrasonic bath. The resulting extracted solutions were measured by HPLC, yielding the amount of CP extracted with PBS.

This kind of recovery with PBS (%) in each extracted specimen was expressed as the ratio between the experimental content of the drug and the expectable value estimated from the preparations of fortified samples.

After the permeation test, membrane specimens were similarly submitted to drug extraction. The calculation of the actual retained amount of drug in each membrane took in account the percentage of recovery obtained in the corresponding preliminary validation.

#### 2.8.3. Drug Retention inside the Permeation Membrane

At the end of the experiment, membranes were removed and carefully cleaned with gauze soaked in a 0.05% solution of sodium lauryl sulphate, washed with distilled water and blotted dry with filter paper. The permeation area was excised and weighted and submitted to extraction as described in Section 2.8.2. The amount of CP retained in each permeation membrane (Q_r_, µg/cm^2^/g) was estimated by correcting the experimental result with the mean value of recovery percentage previously obtained during the validation with fortified standards.

#### 2.8.4. Transmembrane Flux under Sink Conditions

The stationary flux values (Js, µg/h) across the permeation membrane, the permeability coefficient (Kp, cm·h) and the lag time (Tl, h) were calculated at a steady state by linear regression [36].

Results were compared with one-way ANOVA. Differences were considered significant at *p*-value < 0.05.

#### 2.8.5. Histological Integrity of the Permeation Membranes

Associated with the in sensu stricto analytical validation, the integrity of the tissues after the permeation experiment and untreated tissues were investigated with selective staining and optical microscopy. For this, we used ophthalmic and mucous tissue membranes used in diffusion cell permeation tests to investigate the CP recovery (intra-run replicates). In addition, samples of the same size from each untreated tissue (blank histological study) were analyzed to be able to compare treated and untreated samples (*n* = 6 for each biological membrane and *n* = 1 for untreated tissues). Untreated and treated studied tissues were collected and fixed overnight by immersion in 4% paraformaldehyde (PFA) in phosphate buffer (PBS pH 7.4, 20 mM) and further processed for paraffin embedding. Vertical histological sections were placed on a coverslip and stained with hematoxylin and eosin. Subsequently, they were observed and photographed with a Leica DMD 108 light microscope at 400×.

## 3. Results

### 3.1. Analytical Method Validation

Specificity–selectivity: Comparative chromatograms of the matrix effect for vaginal tissue are shown in Figure 3 in association with the microscope photograph of the corresponding membrane specimen. The chromatograms of the other tissues are available as Appendix A.

Linearity and range: Based on the expectable experimental concentrations in the different permeation experiments, two ranges of carprofen concentrations were pre-defined to evaluate linearity (from 100 to 3.125 mg/mL and from 6.25 to 0.78 mg/mL).

Results of detection limit, quantification limit, accuracy and precision from CP calibration curves are summarized in Table 1.

### 3.2. Method Applicability

#### 3.2.1. Permeation Experiment

Figure 4 and Figure 5 show the permeation curves of CP through the ophthalmic and mucous membranes, respectively.

Figure 5 represents permeation curves of CP through buccal, sublingual and vagina mucous membranes.

Permeation parameters for ophthalmic membranes are summarized in Table 2.

Permeation parameters for buccal, sublingual and vaginal mucous membranes are summarized in Table 3.

#### 3.2.2. Drug Recovery and Retention in Membranes

Individual CP amounts recovered with PBS from the different membranes are summarized in Table 4.

Table 5 summarizes the mean amounts of CP retained (Qr) in each of the six membranes.

#### 3.2.3. Histological Integrity of the Permeation Membranes

Histological studies of all the studied tissues have been carried out in order to verify the integrity of studied tissues after the permeation experiment. In all tissues, a blank histological study (without drug) and CP-Sat histological study have been performed.

As we can see in the following histological images of each tissue (Figure 6, Figure 7, Figure 8, Figure 9, Figure 10 and Figure 11), the different layers of the untreated membrane and treated membranes are justified. The mucous membranes are composed of two parts: epithelium (a) and connective tissue (b). Inside the epithelium, the outermost part is the stratified flat keratinized epithelium (a) and the connective tissue is own laminate (b). These two parts are separated by the basal layer. This is the basic structure of the mucous membranes and each tissue presents its own particularities as detailed in the images.

The vaginal mucous tissue has the peculiarity of having many undulations (Figure 6). This shape is to facilitate mounting with the swine male. Another peculiarity of this tissue is that the epithelium (a) may have different thickness depending on the phase of the oestrous cycle.

We can see that the buccal mucosa (Figure 7) presents a very thick epithelium that forms the dermal papilla (c).

The sublingual mucous tissue (Figure 8) presents some peculiarities since it is a more muscular tissue (c) and with collagen fibers (d).

In Figure 9, we can see the cornea. It is a very fragile and fine fabric. The epithelium (a) and the lamina propria (b) are separated by a thin layer called Bowman’s membrane (c).

Figure 10 shows the images of the conjunctiva. It is a highly vascular tissue and we can see hair follicles that belong to the eyelashes (d).

Finally, we can see the structure of the sclera (Figure 11). This tissue is also very fine, fragile and delicate. The episclera (a) is the outermost layer and the stroma (b) is made up of collagen fibers.

## 4. Discussion

### 4.1. Analytical Method Validation

Specificity–selectivity: peaks detected regularly before the first 2 min in all samples were attributed to the sample injection because it appeared regularly in all tested samples (both CCS and also permeation samples). The CP peak appeared between 2–3 min. Resolution calculated for any detected minor peak revealed this method to be acceptably selective for CP [26,27].

Linearity and range: all determination coefficients indicate (statistically) significant differences (*p* < 0.05) of the linear relationships between CP concentration and the chromatographic areas. Satisfactory linearity within the tested concentrations was demonstrated because no statistical differences were found (*p* > 0.05).

Detection limit and quantification limit: predictions from both calibration ranges are similar. Assuming a cell volume of 4.5 mL, and considering that 300 µL× 1500 µg/mL have been used as donor solution, the minimum value of LOQ assures the adequate measurement of up to a minimum of 0.4% of the dose applied in the permeation experiment.

Accuracy and precision from CP calibration curves: both parameters were lower than the 15% limit values suggested in European Medicines Agency (EMA) guidelines [26] and statistical differences (*p* < 0.05) in the ANOVA test were not found between CCS replicates at the same concentration level. These results suggest that the proposed method used is adequately accurate and precise for the entire interval (0.78–100 µg/mL).

### 4.2. Method Applicability

#### 4.2.1. Permeation Experiments

According the permeation experiments, sink conditions [37] were accomplished in the receptor compartment (receptor volume ≥ three times the saturation volume) as a premise for this validation. Sink compliance was verified considering the experimental solubility of CP in PBS and the maximum permeation levels achieved in each case.

Conjunctiva membrane permeation (Figure 4: left) exhibits a typical stationary permeation (linear) since time zero. Conversely, sclera and cornea membranes curves (Figure 4: center and 4: right) are concave, suggesting the existence of a relevant lag time until the second hour (in cornea) and third hour (in sclera). Maximum permeated amount occurred in the case of corneal permeation, and its equivalent concentration did not reach the limit of sink conditions, which is proven by the shape of the curve. Consequently, in the ophthalmic tissues, corneal permeation has the maximum value (49.06 µg), which represents a concentration rounding about 220.78 µg/ mL.

Concerning mucosal membranes, the buccal and sublingual profiles (Figure 5: left and 5: center) are concave, suggesting an increase in the permeation flux after the third hour in both tissues. However, vagina mucous membrane permeation (Figure 5: right) shows linear permeation since the beginning, without a lag time. In this case, the levels at 6 h of the three cases are quite similar, rounding the maximum value at 18.24 µg for the buccal permeation, which represents a concentration of about 218.89 µg/mL.

Summarizing, the maximum permeated amount with ophthalmic tissues is 49.06 µg, which represents 2.80% of the CP saturation value. Relating to the mucosal membranes, the maximum amount permeated is 18.24 µg, which represents 1.02% of the CP saturation value. In both series, the maximum permeated amounts represent less than 20% of the CP solubility, estimated in Section 2.5, which fulfill this experimental premise.

Concerning the likelihood of drug retention values, actual results fall near what has been achieved by other authors with similar drugs. Corneal permeation of NSAIDs is known to be higher than through other ophthalmic structures [38]. The adequacy of ophthalmic tissues of the specifications of variability given by the bioanalysis guidance can be corroborated based on some published comparisons between experimental values and incurred samples, seeming to fall below 20% [39].

After the permeation test, statistical differences between ophthalmic tissues were observed for flux and permeability coefficients. CP flux (Js) is significantly faster through sclera than in conjunctiva and cornea. These results may be due to their different anatomical structures and the product formulation [40,41,42]. In addition, several studies support that, although cornea and sclera have a similar thickness (900µm), the sclera is ten times more permeable [42,43]. No previous studies of the conjunctiva have been found. On the other hand, Kp is significantly higher in sclera than in conjunctiva and cornea. Therefore, conjunctiva and cornea have a lower permeability, CP stays longer in the tissue and this is a beneficial factor for future studies of locally-acting anti-inflammatory treatments with a low number of drug administrations.

On the other hand, flux (Js) is quite low and very similar in the three mucous membranes. Lag time is the time required to reach the steady state [35], suggesting a rapid absorption of CP through buccal and sublingual mucous membranes and more slowly in vagina mucous membrane. These differences can be explained due to the structure of the tissues [44]. No significant differences in Kp values are observed. In summary, permeabilities are lower than in ophthalmic tissues and an acceptable histological innocuousness is expectable.

#### 4.2.2. Drug Recovery and Drug Retention in Membranes

The maximal recovery has been achieved with the vaginal mucous membrane (26.2%) and the lowest with buccal mucous membrane (1.2%).

The use of organic solvents instead of PBS would achieve a higher recovery but would compromise the histological integrity of the membranes, which is the main intention of this experiment. This distortion prevents the performing of the subsequent histological analysis. In fact, previous validation studies [45] prove that CP can be degraded by using alcoholic or halogenated organic solvents, further reinforcing the use of aqueous PBS for extraction testing.

As we can see in Table 5, the retained amount (Qr) in conjunctiva and sclera (22.74 and 48.94 µg/cm^2^/g, respectively) is higher than in cornea (14.21 µg/cm^2^/g). The high retention of CP in the cornea favors a prolonged release. A similar study [46] with pig cornea and CP nanoparticles showed about 50.43 µg/cm^2^/g. Those results are not directly comparable with this experiment due to differences in formulation and methodology, but the similarity of achieved values confirms its suitability.

In studied mucous membranes (see Table 5) Qr is very similar in the vagina and sublingual mucous membranes, with vagina mucous being the highest (26.3 µg/cm^2^/g) in proportional relationship with the transmembrane flux. In contrast, the minor Qr belongs to buccal mucous membrane (10.41 µg/cm^2^/g). These differences between buccal and sublingual mucous may be due to structural tissue differences [44]. In this case, the high Qr in the sublingual mucous makes CP a very interesting candidate drug to be administered locally in this tissue [47].

Similar levels of drug retention in ophthalmic membranes have been obtained with pranoprofen, another NSAID with similar physic-chemical properties [33]. These results confirm that, although the percentage of recovery is pitifully low, the correction calculated on the basis of the validation can be used to estimate the expected total amount of drug retained in each specific tissue.

#### 4.2.3. Histological Integrity of the Permeation Membranes

In order to evaluate the histological images, it is necessary to observe and compare some parameters. The most important part to assess whether a tissue has suffered any damage is the outermost part (in this case the stratified flat keratinized epithelium), since it is the part that has more direct contact with the drug. Furthermore, it is also important to observe that the rest of the cellular structures do not present any type of alteration [48,49]. Therefore, we will analyze the images of each studied tissue separately.

Figure 6, Figure 7 and Figure 8 represent vaginal, buccal and sublingual mucous, respectively. Figure 6 shows the untreated vaginal mucous tissue (Figure 6A) and treated with CP-Sat (Figure 6B). If we compare two figures, we can see no differences are observed and we can say that the outermost part is not damaged in the treated tissue. Figure 7 shows the buccal mucous. As can be seen, part of the epithelium (the outermost part) does not show alterations or cellular changes. The sublingual mucous tissue is represented in Figure 8. It can be seen that both photos (Figure 8A: untreated tissue and Figure 8B: treated tissue) show no damage and therefore that the treated tissue with CP-Sat has not suffered any alteration.

Figure 9, Figure 10 and Figure 11 represent the ocular tissues: cornea, conjunctiva and sclera, respectively. If we look at Figure 9B (cornea treated with CP-Sat), we can see that neither the cell structure nor the epithelium (a) show alterations. Then, we can see Figure 10 (conjunctiva), Figure 10B (conjunctiva treated with CP-Sat) does not present any difference with Figure 10A (untreated fabric). Therefore, no structure has been affected. Finally, as we can see in Figure 11B, no structural part of the sclera is affected.

In summary, we did not observe histopathological damage between the control samples (untreated) and samples treated with CP-Sat. The cellular structure in all the tissues studied shows a normal morphology and distribution. Therefore, we can say that the method used does not affect the cellular structure of the studied tissues.

## 5. Conclusions

In this study, a specific ex vivo procedure and a simple high-performance liquid chromatographic method to quantify CP in permeation and retention samples obtained in porcine mucous membranes or ophthalmic tissues has been set-up and validated. The results demonstrate an adequately selective, accurate and precise method. The validity of the experimental procedure can be confirmed based on the compliance of sink conditions, likelihood of results and histological integrity of the membranes, ensuring its predictability. Results confirm that this experimental set-up is an acceptable procedure to test local distribution of NSAIDs, which is a desirable alternative to circumvent the adverse effects of CP when administered systemically.

## Figures and Tables

**Figure 1 vetsci-07-00152-f001:**
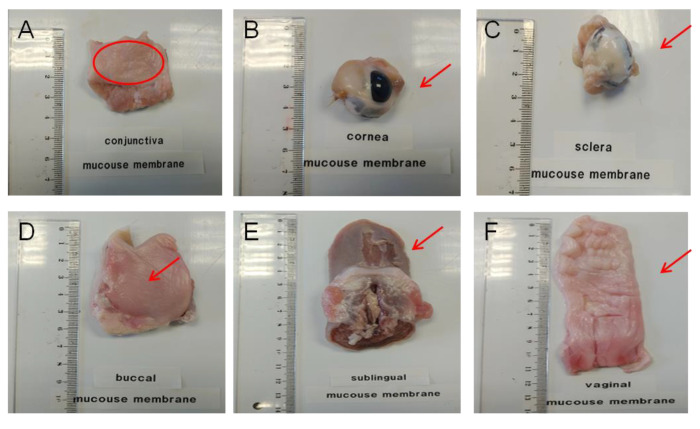
Ex vivo porcine ophthalmic and porcine mucous used as permeation membranes. (**A**): conjunctiva; (**B**): cornea; (**C**): sclera; (**D**): buccal mucous; (**E**): sublingual mucous and (**F**): vaginal mucous.

**Figure 2 vetsci-07-00152-f002:**
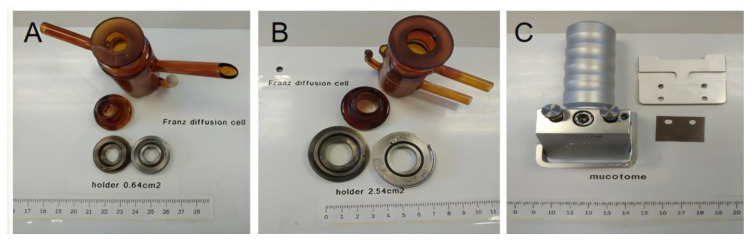
Tools used in permeation tests. (**A**): 0.64 cm^2^ Franz diffusion cell; (**B**): 2.54 cm^2^ Franz diffusion cell and (**C**): mucotome.

**Figure 3 vetsci-07-00152-f003:**
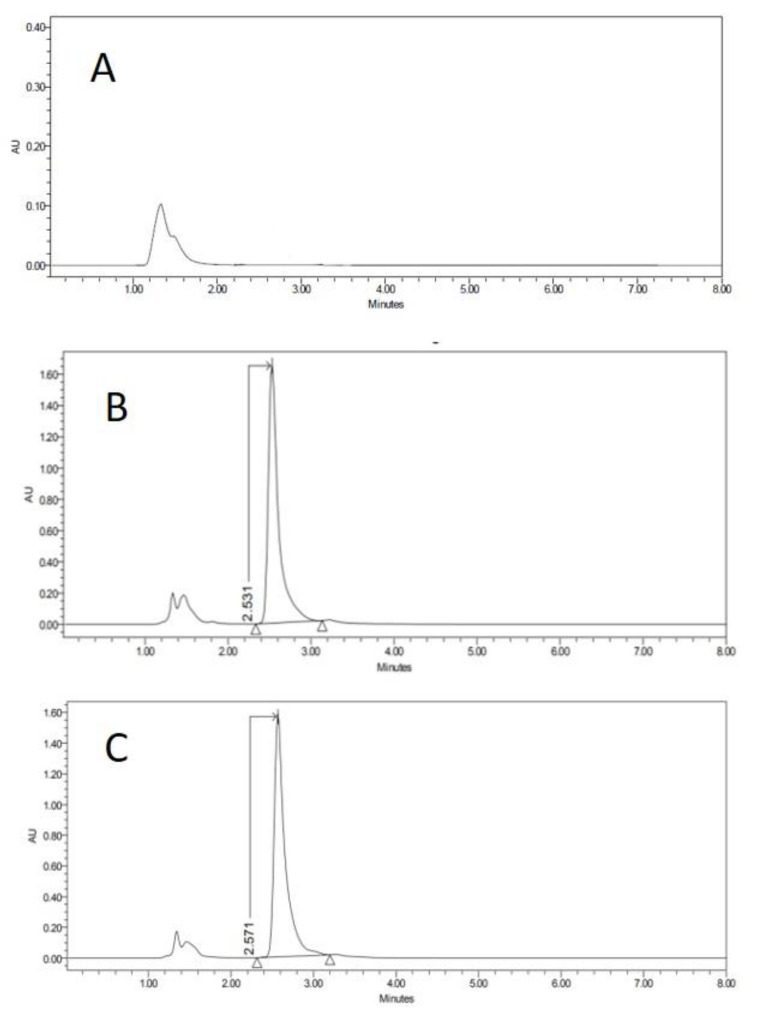
HPLC chromatograms of blank of PBSm (**A**), PBSm spiked with carprofen (**B**) and receptor solution after 6 h of vagina permeation (**C**). Y-axis in the figure is expressed in arbitrary units (AU).

**Figure 4 vetsci-07-00152-f004:**
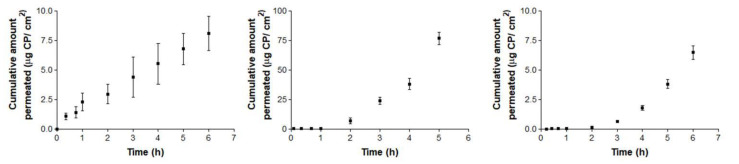
Ex vivo conjunctiva permeation (left), cornea (center) and scleral permeation (right) profiles of carprofen (CP) over 6 h, expressed by mean ± SD of six replicates (*n* = 6).

**Figure 5 vetsci-07-00152-f005:**
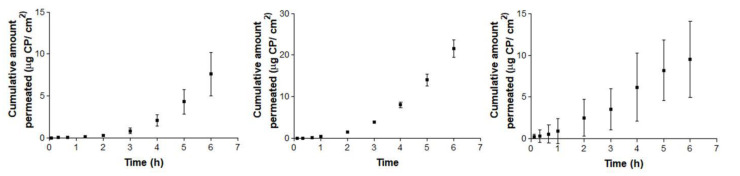
Ex vivo conjunctiva buccal (left), sublingual (center) and vagina permeation (right) profiles of CP over 6 h, expressed by mean ± SD of six replicates (*n* = 6).

**Figure 6 vetsci-07-00152-f006:**
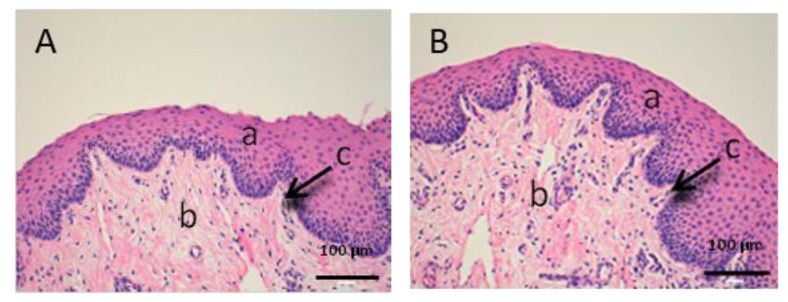
Histological images of vaginal mucous. (**A**): Untreated vaginal mucous observed at 400× and (**B**): Vaginal mucous treated with CP-Sat observed at 400×. a: stratified flat keratinized epithelium; b: own laminate and c: basal layer.

**Figure 7 vetsci-07-00152-f007:**
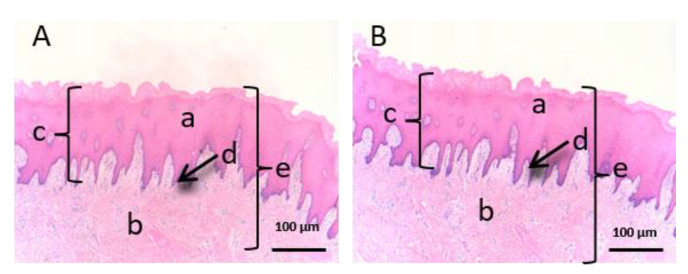
Histological images of buccal mucous. (**A**): Untreated buccal mucous observed at 400x and (**B**): Buccal mucous treated with CP-Sat observed at 400×. a: stratified flat keratinized epithelium; b: own laminate; c: dermal papilla; d: basal layer and e: buccal mucous.

**Figure 8 vetsci-07-00152-f008:**
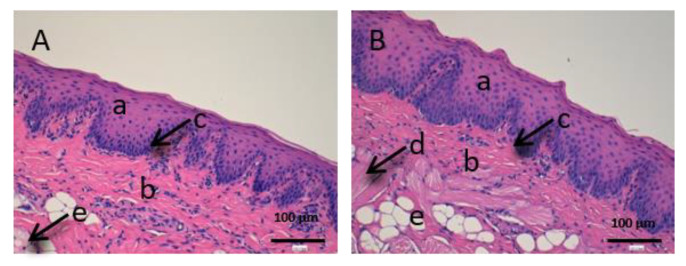
Histological images of sublingual mucous. (**A**): Untreated sublingual mucous observed at 400× and (**B**): Sublingual mucous treated with CP-Sat observed at 400×; a: stratified flat keratinized epithelium; b: own laminate; c: basal layer; d: muscle and e: collagen fibers.

**Figure 9 vetsci-07-00152-f009:**
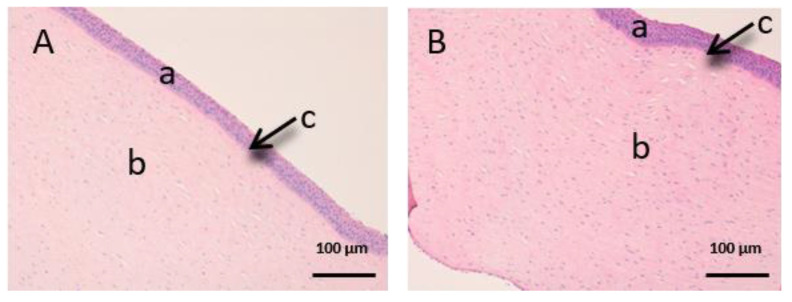
Histological images of cornea. (**A**): Untreated cornea observed at 400× and (**B**): Cornea treated with CP-Sat observed at 400×. a: stratified flat keratinized epithelium; b: own laminate. c: Bowman’s membrane.

**Figure 10 vetsci-07-00152-f010:**
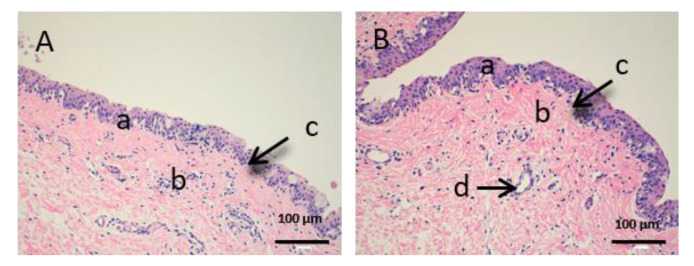
Histological images of conjunctive mucous. (**A**): Untreated conjunctive mucous observed at 400× and (**B**): Conjunctive mucous treated with CP-Sat observed at 400×. a: stratified flat keratinized epithelium; b: own laminate and c: hair follicle.

**Figure 11 vetsci-07-00152-f011:**
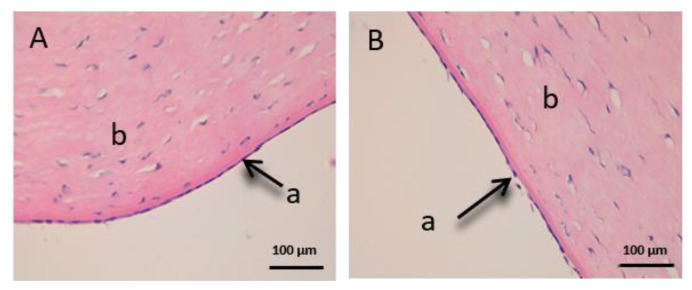
Histological images of sclera. (**A**): Untreated sclera observed at 400× and (**B**): Sclera treated with CP-Sat observed at 400×. a: stratified flat keratinized episclera and b: stroma (collagen fibers).

**Table 1 vetsci-07-00152-t001:** Linearity test of the concentration range of both curves, accuracy and precision of minimum, medium and maximum concentration value and detection and quantification limits of the chromatographic quantification of carprofen.

**Carprofen**	**Linearity Test**	
**Range**	***p*-Value**	
100–3.125 (µg/mL)	0.404	
6.25–0.7813 (µg/mL)	0.0783	
**Min., Medium and Max. Concentrations**	**Accuracy RE (%)**	**Precision RSD (%)**
100 (µg/mL)	0.21	0.60
12.5 (µg/mL)	0.96	1.77
0.7813 (µg/mL)	0.87	1.95
	**Mean**	**Desvest.**
**Detection limit (µg/mL)**	0.17	0.13
**Quantification limit (µg/mL)**	0.51	0.40

**Table 2 vetsci-07-00152-t002:** Median, maximum and minimum values of flux (Js), lag time (Tl) and permeability coefficient (Kp) of CP at 6 h from CP saturated solution (CP-Sat) through ophthalmic tissues (conjunctiva, cornea and sclera).

Carprofen	Conjunctiva (CJ)	Sclera (SC)	Cornea (CO)	*p*-Value
Js (µg/h)	0.22	26.54	1.85	SC vs. CO **
	(0.09–0.34)	(15.52–37.56)	(0.14–2.34)	SC vs. CJ ***
Tl (h)	0.91	1.57	3.29	NS
	(0.58–1.23)	(1.52–1.61)	(2.17–3.34)	
Kp × 10^2^ (cm·h)	0.045	5.53	0.39	SC vs. CO *
	(0.019–0.071)	(3.23–7.82)	(0.03–0.49)	SC vs. CJ ***

* *p*-value < 0.05; ** *p*-value < 0.01; *** *p*-value < 0.00.

**Table 3 vetsci-07-00152-t003:** Median, maximum and minimum values of flux (Js), lag time (Tl) and permeability coefficient (Kp) of CP at 6 h from CP-Sat through mucous membranes (buccal, sublingual and vaginal).

Carprofen	Buccal (B)	Sublingual (SB)	Vagina (V)	*p*-Value
Js (µg/h)	1.61	5.1	7.59	NS
	(0.46–2.77)	(2.38–6.76)	(1.68–13.51)	
Tl (h)	3.27	2.83	0.13	B vs. V ***
	(3.22–3.31)	(0.86–3.47)	(0.02–0.23)	SB vs. V *
Kp × 10^2^ (cm·h)	0.085	1.0625	1.58	NS
	(0.024–0.45)	(0.5–1.14)	(0.35–2.81)	

* *p*-value < 0.05 and *** *p*-value < 0.001 and NS: Not significant.

**Table 4 vetsci-07-00152-t004:** Results of the carprofen recovery with phosphate buffered saline (PBS) of the battery of fortified samples expressed by amount before and after penetration; extracted amount of tissues and final recovery % with relative standard derivation (RSD).

Membrane	Sample	Amount in Solution, before (µg)	Amount in Solution, after (µg)	Charge (µg)	Extracted Amount (µg)	Recovery (%)	RSD (%)
Conjunctiva	FSCN-1	91.67	0.79	90.88	16.14	17.8%	15.93
	FSCN-2	91.67	0.73	90.94	12.88	14.2%	
Sclera	FSSC-1	91.67	47.70	43.97	1.05	2.4%	7.07
	FSSC-2	91.67	51.51	40.16	1.06	2.6%	
Cornea	FSCR-1	91.67	51.51	40.16	1.22	3.0%	14.17
	FSCR-2	91.67	49.13	42.54	1.58	3.7%	
Buccal	FSBC-1	98.54	0.35	98.19	1.44	1.5%	15.44
	FSBC-2	98.54	0.90	97.64	1.15	1.2%	
Sublingual	FSSL-1	98.54	56.44	42.1	1.12	2.7%	20.06
	FSSL-2	98.54	45.02	53.52	1.07	2.0%	
Vaginal	FSVG-1	98.54	1.28	97.26	23.55	24.2%	5.56
	FSVG-2	98.54	0.85	97.69	25.59	26.2%	

**Table 5 vetsci-07-00152-t005:** Median, maximum and minimum values of retained amount (Qr) of carprofen at 6 h from CP-Sat through ophthalmic tissues (conjunctiva, cornea and sclera) and mucous membranes (buccal, sublingual and vaginal).

**Carprofen**	**Conjunctiva (CJ)**	**Cornea (CO)**	**Sclera (SC)**	***p*-Value**
Qr (µg/cm^2^/g)	22.74	14.21	48.94	SC vs. CO **
	(19.04–26.45)	(12.38–16.05)	(41.71–56.17)	
**Carprofen**	**Buccal (B)**	**Sublingual (SB)**	**Vagina (V)**	***p*** **-Value**
Qr (µg/cm^2^/g)	10.41	18.58	26.3	B vs. V **
	(9.22–11.6)	(18.25–18.91)	(26.01–26.59)	

** *p*-value < 0.01.

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
