# Peer review of "Carprofen Permeation Test through Porcine Ex Vivo Mucous Membranes and Ophthalmic Tissues for Tolerability Assessments: Validation and Histological Study"

_vetsci, 2020, doi:10.3390/vetsci7040152_

Round 1

Reviewer 1 Report

In the title it is written 'histological study', but I did not find any information as regard the histological findings in results section and the histological findings was not discussed. I suggest to change the title of the manuscript or to insert an histological section in both results and discussion sections

Line 85: I would like to see the protocol number of the ethic committee 

Line 89: are you sure that freezing the tissue at -20°C do not alter the permeability? 

Author Response

Please see the attachment, thanks

Reviewer 2 Report

The manuscript is well written. I’ve comments related to validation results which needs

The manuscript is well written. I’ve comments related to validation results which needs to be elaborated and discussed in details to support conclusions

  • Figure 4 A: Why the CP peak (Rt 2-3min) is showing up in PBS blank sample
  • Line 321: The author must add the linearity curves for three interday runs
  • Tbale 1: The author must elaborate and explain table1. What parameters has been assessed to calculate %RSD? Which samples belong to what day?

Author Response

Please see the attachment, thanks

Reviewer 3 Report

The manuscript Carprofen permeation test through porcine ex vivo mucous membranes and ophthalmic tissues for tolerability assessments: Validation and histological study it's quite interesting and proposed and validates a new methodology. I believe that it brings new information, was made in solid science, and can be published after corrections. 

My major issue is about the discussion, it´s not a proper discussion with much of the repetition of the results. Therefore the discussion section must be entirely revised. 

Some additional comments:

L20: anti-inflammatoryactivity

There is a missing space

L23: vagina

I think vaginal would be more appropriate considering the sentence.

Regarding the supplementary material, please avoid the use of uncommon abbreviations in the figure legends (e.g. CP). Please define AU from the Y-axis in the figure legend.

L27: correlation coefficients (R2 ˃0.998 and R2 ˃0.999)

R2 stands for the determination coefficient. R is the correlation coefficient

L30: Recovery levels of drug in tissue samples were assessed with aqueous PBS buffer to preserve the histological integrity.

This is methods but is placed after some result. Please move up to your methods part of the abstract.

L33:  As a proof of concept, a series of CP permeation tests in vertical Franz  diffusion cells has been performed followed by a histological study for critical evaluation. Furthermore, synthetic tissue retention-like samples were prepared to verify the value of this experimental study.

Why these methods informations are in the result part of your abstract. I understand that sometimes your results lead to do something and this could be the arrangement, but should be important to justify why you did this. It was because of your results? If you had planned to do a proof of concept in the beginning of the study this information should be moved up in the abstract.

The abstract is a little bit confusing, Readers not quite familiar with the field will not understand the study reading the abstract. I suggest more clear information at this part: Main aim of this paper is to validate the suitability of ex vivo permeation experiments of CP with porcine mucous membranes (buccal, sublingual and vagina) and ophtalmic tissues (cornea, sclera and conjunctiva) intended to be representative of naïve in vivo conditions.

Perhaps more information on ex vivo and permeation. This suggestion is not mandatory, it is just only if you intend to attract more readers not quite familiar to the filed.

L55: On the whole,

Poor writing, please revise.

L46: It is used in veterinary majoritarily as anti-inflammatory but is withdrawn from humans due to episodic adverse effects such as photoallergic cases after bad manipulation at industrial scale.

Poor writing, please revise. Start saying about the human situation and then inform that it´s still being used in veterinary medicine…

L69:  As a proof of concept of this preliminar validation, the intrinsic permeation of CP through these tissues was investigated to evaluate the adequacy of this ex vivo test to estimate the corresponding parameters of drug permeation-penetration. In addition, histological examination of the tissues used  in validation and permeation runs have been carried out to confirm that CP does not affect or damage the cell structure and can be formulated for local administration.

Too much method´s section information in the introduction. You don´t need to specify the tissues, you don´t need to specify the histological analysis. Please revise.

Methods, please use this standard in all your methods description: (Fisher Scientific, Leicester shire, UK).

L84-85: Please be clear about the approval of your study and inform the approval/protocol number.

Please revise the figures 1 and 2 as one panel, put the photos together for a better presentation, and put the corresponding letter into the figures: A to F;

L121: please add a supplementary file with all the results (mean, DP, CV) of the calibration procedures and quality control used including linearity, range, accuracy, recovery, and precision

L121: FS*

Why the *?

L144: (Waters LCM1 plus (Waters Co., Milford, Ma, USA)

Please revise.

L216: Main premises to be fulfilled were:

Please remove the paragraph and marks and put this in one sentence. (1 Sink conditions across the transmembrane; (2) Likelihood of the drug retention levels; (3) Histological integrity of the membranes after the experiment.

Figure 3A, please remove space between figures and enlarge each figure. Also put the letter indication each one is A, B or C.

You described this software differently, please be consistent:

Graph Pad Prism® software version 5.01 (GraphPad Software Inc., San Diego, CA, USA)

ANOVA analysis (Prism® , v. 5.00, GraphPad Software Inc., San Diego, CA, USA)

ANOVA as in 2.7.6.

I don´t think you need to repat the software. Inform once ate some point that all statistical analysis was done using the software. You don´t need to refer to another topic when doing ANOVA, since it’s a quite simple method.

Figure 4. There is a missing photo.

In figure 4, what the 3 separated figures stand for?? Why you have 3 results when the figure legend it is not mentioned.

Do you really need to separate all the analytical validation into several subtopics, with some with only one phrase? Such as:

3.1.3. Determination limits

 Results are summarized in Table 1. 

 3.1.4. Accuracy and precison from CP calibration curves

 Results are summarized in Table 1.

You can put all your information on one topic of quality control and remove all the subtopics 93.1.1 to 3.1.5).

Check the word precision, it´s a typo.

Tables, remove horizontal lines.

L280: Individual CP amounts recovered with PBS from the different membranes (see section 2.7.5)

Please remove this: (see section 2.7.5)

From the entire manuscript, you don´t need it.

Please define abbreviation in the footnotes of the table.  

I suggest you remove subtopics 3.2.1 to 3.2.3 into one topic.

Please remove the subtopics of your discussion. Or at least leave only 2.

The discussion part from 4.1.1 to 4.1.5 it´s not a discussion, it´s a repetition of the results. Remove it and discussed your work properly. Just say that your method validation was ok and compare with some other similar studies and methods validation.

You don´t need to make a reference to each result in the discussion section. Please revise this.

Again, the discussion is too much repetition of the results. With the number presented. Please revise your entire discussion and comment about the meaning of your results, not put your results again. How your results are important, how they can be compared. I understand that maybe you don´t have much to discuss since is a methodology article, so just remove all the duplicated results in the discussion.

Author Response

Please see the attachment, thanks

Round 2

Reviewer 1 Report

Thanks to the authors for adding the informations requested in previous revision.

I have only few concerns.

At lines 88-89 I think a verb is lacking. 

Lines 489-519: these are results of the histology rather than the discussion of the results. I suggest to move this section in results section and to discuss better the histology results.

Author Response

Please, see the attached document

Reviewer 2 Report

The authors have addressed the comments and can be accepted in present form.

Author Response

Please, read the attached document, thanks

Reviewer 3 Report

The authors successfully corrected all my suggestions and provided a detailed revision note with specific responses to all my remarks. I believe that this manuscript is quite interesting, proposed, and validates a new methodology with very detailed validation results and adequate analytical procedures. The work brings new information, made through robust science, and can be published in the journal. 

I have one suggestion, the figures are ugly (poor presentation). Please correct all figures according to the example: https://doi.org/10.1371/journal.pone.0177544 or https://doi.org/10.1371/journal.pone.0192270

Figures must be closest to each other without excessive blank spaces (forming a panel). The distance between all figures (blank space) must have exactly the same size in all figures. This manuscript will be permanently evaluable until de end of times, the authors must assure good presentation and consistency in figures.  

This seems odd to me: Figure 10: Photo 9:

Please change for letters, Figure 10A, 10B. Figure 9A, 9B. This is the most used form and improves the quality of the figures. Also, when removing the number to replace by letter, you can reduce the size of the letters. 

Figure 11, photo 12 appears to be smaller the photo 11, please make sure that all photos have the same size.  

Author Response

(The authors gave the same response as above.)
